# Predicting preterm birth using explainable machine learning in a prospective cohort of nulliparous and multiparous pregnant women

Wasif Khan[1,2], Nazar Zaki[1,2], Nadirah Ghenimi[3]*, Amir Ahmad[1,4], Jiang Bian[5], Mohammad M. Masud[1,2,4], Nasloon Ali[6], Romona Govender[3], Luai A. Ahmed[4,6]

1 Department of Computer Science and Software Engineering, College of Information Technology, United Arab Emirates University, Al Ain, UAE, 2 Department of Information Systems and Security, College of Information Technology, United Arab Emirates University, Al Ain, UAE, 3 Department Family Medicine, College of Medicine and Health Sciences, United Arab Emirates University, Al Ain, UAE, 4 Zayed Centre for Health Sciences, United Arab Emirates University, Al Ain, UAE, 5 Department of Health Outcomes and Biomedical Informatics, College of Medicine, University of Florida, Gainesville, Florida, United States of America, 6 Institute of Public Health, College of Medicine and Health Sciences, United Arab Emirates University, Al Ain, UAE

* nghenimi@uaeu.ac.ae

**Data Availability Statement:** Relevant data are available within the paper. Due to ethical restrictions, the complete dataset cannot be

## Abstract

Preterm birth (PTB) presents a complex challenge in pregnancy, often leading to significant perinatal and long-term morbidities. "While machine learning (ML) algorithms have shown promise in PTB prediction, the lack of interpretability in existing models hinders their clinical utility. This study aimed to predict PTB in a pregnant population using ML models, identify the key risk factors associated with PTB through the SHapley Additive exPlanations (SHAP) algorithm, and provide comprehensive explanations for these predictions to assist clinicians in providing appropriate care. This study analyzed a dataset of 3509 pregnant women in the United Arab Emirates and selected 35 risk factors associated with PTB based on the existing medical and artificial intelligence literature. Six ML algorithms were tested, wherein the XGBoost model exhibited the best performance, with an area under the operator receiving curves of 0.735 and 0.723 for parous and nulliparous women, respectively. The SHAP feature attribution framework was employed to identify the most significant risk factors linked to PTB. Additionally, individual patient analysis was performed using the SHAP and the local interpretable model-agnostic explanation algorithms (LIME). The overall incidence of PTB was 11.23% (11 and 12.1% in parous and nulliparous women, respectively). The main risk factors associated with PTB in parous women are previous PTB, previous cesarean section, preeclampsia during pregnancy, and maternal age. In nulliparous women, body mass index at delivery, maternal age, and the presence of amniotic infection were the most relevant risk factors. The trained ML prediction model developed in this study holds promise as a valuable screening tool for predicting PTB within this specific population. Furthermore, SHAP and LIME analyses can assist clinicians in understanding the individualized impact of each risk factor on their patients and provide appropriate care to reduce morbidity and mortality related to PTB.

publicly available as the data contains potentially sensitive patient information. Data would be available upon justified request from the Mutaba'ah Study (mutabaah@uaeu.ac.ae) after ethical approval.

**Funding:** This work was supported by a grant from Zayed Center for Health Sciences, United Arab Emirates University (31R239).

**Competing interests:** The authors have declared that no competing interests exist.

## Introduction

The World Health Organization reports that, every year, approximately one in ten babies is born prematurely, before completing 37 weeks of gestation [1]. Preterm birth (PTB) complications rank as a leading cause of death among children under the age of five, with an estimated 75% of the one million PTB-related deaths being preventable [1]. The incidence of PTB varies across 184 countries, ranging from 5 to 18% [2]. In 2019, the United States (US) reported a PTB prevalence of roughly 10.2% in 2019, while the United Arab Emirates (UAE) estimated a prevalence of around 6.3% in the same year, considering both Emirati and expatriate populations [3]. PTB is a complex condition with multifactorial causes [1]. Among the widely studied risk factors are maternal demographics and characteristics (such as advanced maternal age and adolescent pregnancy, social determinants (including smoking and substance use), economic factors, medical complications, obstetric history (as short interpretation interval and previous PTB), and conditions specific to the current pregnancy [1–3]. However, from the extensive list of risk factors, predicting the occurrence of PTB is challenging because the signs and symptoms of preterm labor are common and can be nonspecific. Thus, the current assessment methods for predicting individual PTB risk are problematic, particularly for nulliparous women with no obstetric history. Traditionally, these statistical models rely on single factors, such as demographic history, obstetric history, and clinical characteristics. Machine learning (ML)-based models have successfully predicted the risks of numerous medical conditions [4]. Several studies have employed ML models to predict PTB. However, despite their mathematical sophistication, these models are "black-boxes," lacking both interpretability and explanation. Therefore, for clinical utility, the predictions made by ML models must be interpretable by clinicians, enabling them to assess PTB risk for each patient and understand the contribution of individual risk factors to these predictions [5, 6]. Explainable ML models, such as the SHapley Additive explanations (SHAP) and local interpretable model-agnostic explanations (LIME), can be used to achieve this goal. Once clinicians understand the ML model results, they gain confidence in the ML model prediction of impending PTB risk [7–11]. This informed clinical risk stratification can substantially improve the health outcomes of preterm infants and their mothers. Consequently, this study aimed to predict PTB in nulliparous and parous women using ML models, identify important predictors associated with PTB, and provide explanations for the contribution of each risk factor to PTB prediction using SHAP and LIME.

## Materials and methods

### Data and population

The dataset utilized in this analysis was obtained from an ongoing prospective maternal and child cohort study, the Mutaba'ah Study, conducted in Al Ain, UAE [12]. Eligible participants included all pregnant women aged 18 y and above from the Emirati population residing in Al Ain City, who provided informed consent for themselves and their newborns. This study received approval from the Abu Dhabi Health Research and Technology Ethics Committee (DOH/CVDC/2022/72) and was conducted in strict accordance with the Declaration of Helsinki. Prior to data collection, written informed consent was obtained from all participants. This analysis encompassed 3509 women with singleton pregnancies recruited between May 2017 and February 2021.

As certain risk factors did not apply to nulliparous women, such as previous PTB, previous cesarean section (CS), and parity, the sample was separated into nulliparous mothers (n = 801) and parous mothers (n = 2708).

Participants were followed up during pregnancy, with data collection conducted through self-administered questionnaires and medical records. PTB was categorized based on gestational age at birth, provided in weeks. Any birth occurring before 37 weeks of gestation was classified as PTB. Clinicians performed a scoping review of the literature on PTB risk factors. Simultaneously, a review of the features employed in existing PTB prediction studies utilizing ML models was carried out, as shown in S1 Table. A set of risk factors was derived from the common factors present in both current ML-based studies and medical literature. This combined list was then filtered using data from the electronic medical records (applying ICD 10 coding) and relevant information from the questionnaire (S2 Table). Notably, 35 relevant features were identified in this study (S2 Table) related to ICD10 codes.

## Study population characteristics

Descriptive statistics were employed to illustrate and compare the distribution of the characteristics of the study population based on PTB status. Continuous variables were represented by means and standard deviations, discrete quantitative variables by medians and ranges, and categorical variables by counts and percentages. Student's t-test was used to determine the differences between group means for continuous variables, while categorical variables were compared using Pearson's Chi-square test or Fisher's exact test. Statistical analyses were performed using Stata 16.1 (Stata Corp, College Station, TX, USA). A p-value less than or equal to 0.05 was considered statistically significant.

## Machine learning models

We used a combination of domain knowledge and empirical evaluations to select the models. Specifically, we chose models that have been widely employed in the literature with demonstrated good performance in similar studies (Table 1).

**Literature review.** In this study, the performances of six ML classifiers were evaluated to select the most accurate method for predicting PTB in the population. These ML models include the support vector machine (SVM) [32], random forest (RF) [33], logistic regression (LR) [34, 35], multilayer perceptron (MLP) [36], gradient boosting machine (GBM) [37], and XGBoost [38]. While XGBoost outputs variable importance, it does not measure the direction and level of impact of the variables on the outcomes.

**SHapley Additive exPlanations (SHAP).** SHAP values were introduced to better explain the contribution of features or risk factors to the outcome, specifically PTB [39–41]. The SHAP Shapley values are an attribution method that fairly assigns predictions to individual features. SHAP is a computational method for calculating Shapley values, which also suggests global interpretation methods based on combinations of Shapley values across the dataset. A higher SHAP value indicates that a feature increases the likelihood of PTB, while a lower SHAP value suggests that a feature reduces the outcome likelihood. Thus, the SHAP method can rank the importance of features and reveal the relationship between these features and the outcome. Further details regarding SHAP [5, 19–22] are provided in the S1 File. We obtained a list of the ten most important risk factors and their SHAP values (refer to the Results section).

**Local interpretable model-agnostic explanations (LIME).** LIMEs offer explanations for predictions by replacing a complex model with a locally interpretable surrogate model [42]. Therefore, we performed a risk factor-based analysis of individual patients using LIME. Further details regarding the LIME are included in the S1 File.

**Table 1. Related research on PTB prediction using machine learning models.**

| Reference | Problem and approach | Methods used | Performance |
|---|---|---|---|
| Mercer et al. [13] | PTB prediction using statistical analysis | LR | The authors performed univariate and multivariate LR to identify the risk factors and odds ratios for PTBs. |
| Lee et al. [14] | PTB Classification | ANN, LR, DT, NB, RF, and SVM. | ANN, LR, DT, NB, RF, and SVM achieved accuracies of 0.911, 0.918, 0.832, 0.111, 0.891, and 0.914, respectively. BMI and hypertension were the most important features. |
| Tran et al. [15] | PTB Classification | SSLR and RGB classifier. | SSLR achieved AUC of 0.85 and 0.79 for predicting PTB in the 34th and 37th week of gestation, respectively. RGB achieved AUC of 0.86 and 0.81 for predicting PTB in the 34th and 37th week, respectively. |
| Taha et al. [3] | LBW and PTB analysis | | Association between PTB and LBW was found in the UAE population. |
| Sun et al. [16] | PTB Classification | RF, SVM, ANN, k-means, and NB. | PTB prediction at different gestational periods, including 20th, 22nd, 24th, and 26th weeks. RF achieved better performance with AUC, sensitivity, and specificity of 0.89, 0.75, and 0.88, respectively in the 26th week of gestation. Twenty variables were considered important. |
| Koivu et al. [17] | PTB Classification | Only complete samples, correlation analysis, ANN, LR, and LightGBM. | An AUC of 0.67 was achieved for CDC data, whereas a maximum AUC of 0.64 was achieved for the NYC dataset via ANN and Light GBM. |
| Raja et al. [18] | PTB Classification | Feature selection, SVM, RF, DT, and LR. | SVM achieved an accuracy, sensitivity, and specificity of 0.90, 0.89, and 0.78, respectively. |
| Belaghi et al. [19] | PTB Classification | PTB in nulliparous women during the first and second trimester, RF, LR, DT, and ANN. | Sensitivity and specificity of LR were 0.50 and 0.64 in the first trimester, respectively, whereas they were 0.29 and 0.84 for RF, respectively. Similarly, in the second trimester, the sensitivity and specificity of RF were 0.45 and 0.94, respectively, and those of ANN were 0.62 and 0.84, respectively. |
| Belaghi et al. [20] | PTB prediction | | AUC of LR in the first trimester were 0.68 and 0.73 for nulliparous and multiparous women, respectively, whereas in the second trimester, they were 0.72 and 0.78, respectively. |
| Diaz et al. [21] | PTB Classification | Missing data imputation using KNN, noise reduction techniques. DT, C5.0, NNet, KNN and RF. | C5.0 classifier with robust noise filter achieved sensitivity, specificity and F-score of 0.86, 0.78, and 0.85, respectively. |
| Lee et al. [22] | PTB prediction | LR, ANN, and RF. | AUC within the range of 0.52–0.58 was achieved for a highly imbalanced dataset. |
| Guang et al. [23] | PTB Classification | SVM, RF, ANN, and LR. | AUC, specificity, and sensitivity of 0.92, 0.94, and 0.78, respectively were achieved. |
| Cately et al. [24] | PTB Classification | Data resampling and ANN. | AUC of 0.71 was achieved with sensitivity of 0.33. |
| Khatibi et al. [25] | PTB Classification using MapReduce | MapReduce of feature selection, missing data imputation, DT, RF, SVM and ensemble | AUC of 0.68 with important risk factors identification. |
| Li et al. [26] | PTB and PM 2.5 relationship | statistical analysis to identify relationship between PM of 2.5 and PTB. | PTB mother had exposure to PM of 2.5 |
| Rittenhouse et al. [27] | PTB prediction | Super learner model, RF, and LR. | AUC of 0.97 whereas Positive Predictive Value was only 0.53. |
| Chen et al. [28] | PTB prediction | DT and NN. | 15 crucial risk factors were identified. |
| Prema et al. [29] | PTB classification in diabetic mothers | SMOTE for data balancing, SVM, and LR. | Maximum F-score of 0.80 using SVM. |
| Moreira et al. [30] | PTB and APGAR score prediction using SVM | SVM with smooth linear kernel. | ROC of 0.78 and FR rate of 0.26. |
| Aung et al. [31] | Biomarkers for PTB prediction | LR, RF, and elastic net. | AUC of 0.84 using RF classifier. |

PTB: Preterm birth; LR: logistic regression; ANN: artificial neural network; DT: decision tree; NB: naïve Bayes; RF: random forest; SVM: support vector machine; BMI: body mass index; SSLR: stabilized sparse logistic regression; RGB: randomized gradient boosting; AUC: area under the ROC curve; LBW: low birth weight; UAE: United Arab Emirates; CDC: Center for Disease Control and Prevention; Light GBM: light gradient boosting machine; NN: neural network, PM: particular matter; ROC: receiver operating characteristic, SMOTE: synthetic minority oversampling technique, APGAR: appearance pulse grimace activity and respiration; and FR: fertility rate.

## Experimental settings

The experiments were conducted as follows: First, the missing values in the dataset are replaced with a missed-forest imputation algorithm [43].

The algorithm, known as Miss Forest, was used to handle the missing values in the datasets. It utilizes the RF algorithm, which is an ML technique. Miss Forest treats missing values as a distinct category and predicts them using other variables in the dataset. It iteratively imputes missing values by creating an RF model and refining the predictions in subsequent iterations. This approach allows the algorithm to capture the complex relationships within the data. Subsequently, a tenfold cross-validation was conducted on the dataset to evaluate the average performance. We divided the data into two sets: a training set, which contained 80% of the data, and a test set, encompassing the remaining 20%. This partition ensured that both the training and testing sets included the same proportion of preterm samples. The six most commonly used ML classifiers were employed to identify the best classifier for the task. The evaluation criterion of the area under the Receiver Operating Characteristic (ROC) curve (AUC) was used. The AUC represents the performance of a classifier in distinguishing between positive and negative instances. A higher AUC value indicates a superior classification performance. The ROC curve with a good predictive performance exhibited an AUC close to 1. Thereafter, the global behavior of the best ML model was explained to identify the risk factors using SHAP. A higher SHAP value signifies that the feature increases the likelihood of PTB, while a lower SHAP value suggests that the feature reduces this likelihood. Finally, individual SHAP patient analyses were conducted to identify the underlying risk factors associated with each patient. For comparison, the LIME was used for individual patient analyses. All experiments were conducted using Python 3.8 on a personal computer with an Intel (R) Core i9-9900 CPU@ 3.10 GHz and 8 GB RAM. Fig 1 illustrates the entire methodology, from data collection and preprocessing, the application of different ML algorithms to select the best-performing ML model, identification of the important risk factors and their relative risk scores for PTB, and finally, the individual patient analyses and clinical recommendations.

## Results

### Study population characteristics

The distribution of risk factors and descriptive characteristics of the parous (n = 2708) and nulliparous (n = 801) mothers are presented in Table 2 and S1 Table, respectively. The overall incidence of PTB in this study was 11.23% (11% in parous women and 12.1% in nulliparous women).

In parous mothers (Table 2), low levels of education, exposure to passive smoking, and history of infertility treatment were more frequent among women with PTB than among those without PTB. Parous women with PTB exhibited substantial differences from those without PTB in terms of maternal age, gravidity, preexisting hypertension, preexisting diabetes mellitus, previous PTB, previous cesarean delivery, or previous pregnancy loss. During pregnancy, a considerable proportion of parous mothers with PTB suffer from conditions such as preeclampsia, antepartum hemorrhage, oligohydramnios, infection of the amniotic sac, placenta previa and placental disorders, Streptococcus carrier B, or genitourinary infection. The most significant characteristics for women with PTB were higher maternal age, higher gravidity, and a low level of education. Exposure to passive smoking, history of infertility treatment, preexisting hypertension, preexisting diabetes mellitus, and history of PTB were more frequent in women with PTB (Table 2).

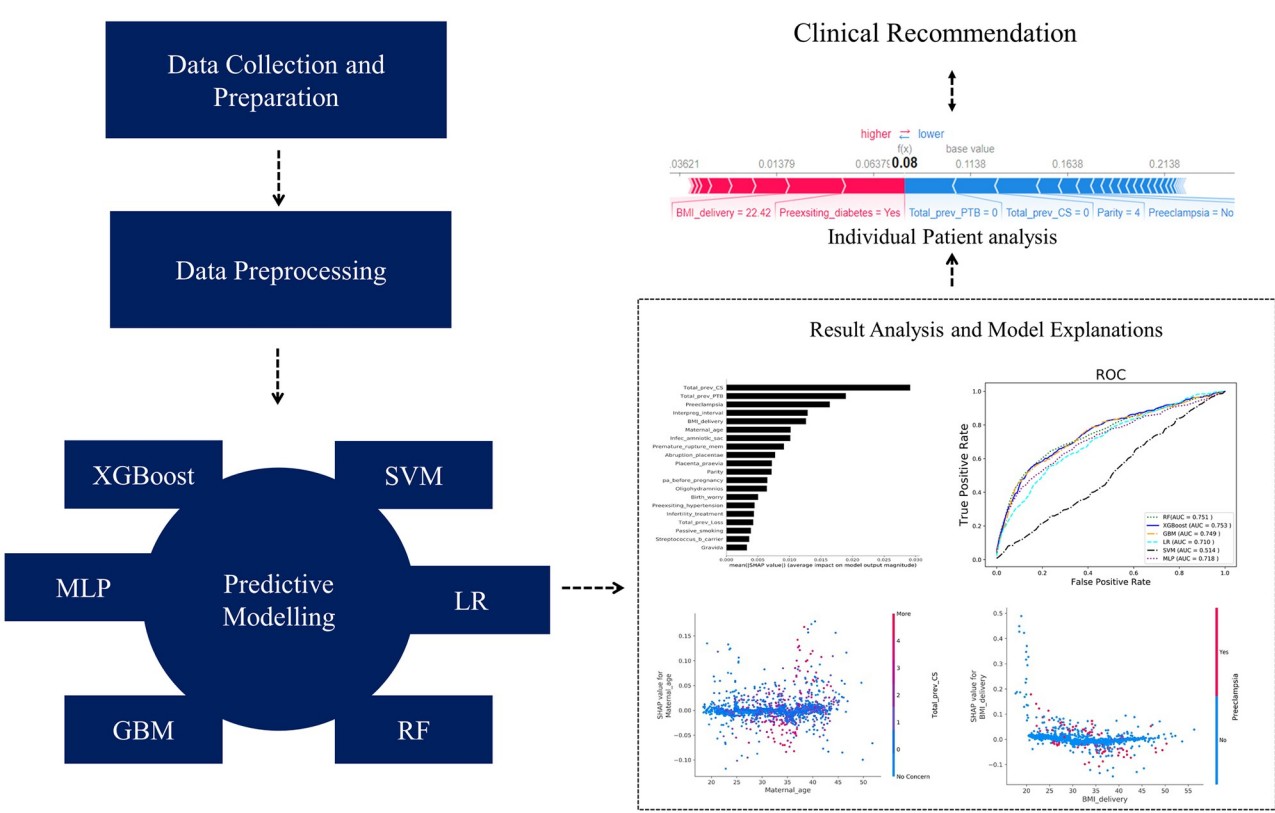

**Fig 1. Proposed methodology for predicting PTB using machine learning models.**

Among nulliparous mothers (Table 3), significant differences were observed in the self-reported planning status of pregnancy, physical activity before pregnancy, and history of infertility treatment between those with and without PTB. For pregnancy and delivery characteristics, preexisting diabetes mellitus, preeclampsia, antepartum hemorrhage, oligohydramnios, infection of the amniotic sac, premature rupture of membranes, and placental abruption were considerably more common in nulliparous mothers with PTB than in those without PTB.

## Performance and interpretation of the ML model

**Parous women.** For parous women, the LR, SVM, MLP, GBM, RF, and XGBoost models achieved an AUC of 0.720, 0.521, 0.706, 0.720, 0.726, and 0.735, respectively (Fig 2A). As XGBoost exhibited the highest AUC, signifying the best predictive performance among these models, we employed it for explainable analysis using SHAP and LIME.

The SHAP plot based on the weights of the risk factors is depicted in Fig 2B. This figure showed that the most important risk factor was a history of PTB, followed by a history of CS, and preeclampsia in the parous population. Other important risk factors include maternal age, placenta previa, BMI at delivery, and interpregnancy interval. A summary graph of the ten most important risk factors is illustrated in Fig 2C. The prediction of a set of patients using SHAP is depicted in Fig 3. The first patient (parous mother), represented in Fig 3(a), was at an exceptionally low risk of PTB delivery (0.0). This low risk is primarily attributed to factors such as the absence of a history of PTB, a maternal age of 28 y, and long interpregnancy interval (903 d). The data of the second patient presented in Fig 3(b) demonstrated a medium risk

**Table 2. Descriptive characteristics of the parous pregnant women.**

| Risk Factors | Total parous pregnant women | PTB | No PTB | *p*-value |
|---|---|---|---|---|
| **Total** | 2708 | 297 | 2411 | |
| **Self-reported characteristics** | | | | |
| Planned Pregnancy | 1337 (52.5) | 131 (9.8) | 1206 (90.2) | 0.056 |
| Employment | 824 (32.8) | 101 (12.3) | 723 (87.7) | 0.149 |
| Education >12 y | 1187 (47.3) | 115 (9.7) | 1072 (90.3) | 0.046 |
| Consanguinity | 643 (45.4) | 65 (10.1) | 578 (89.9) | 0.736 |
| Passive Smoking | 825 (32.4) | 106 (12.9) | 719 (87.1) | 0.031 |
| Physical activity before pregnancy | | | | |
| *Never* | 1317 | 153 (11.6) | 1164 (88.4) | 0.782 |
| *1–2 times/week* | 434 | 43 (9.9) | 391 (90.1) | |
| *3–5 times/week* | 366 | 42 (11.5) | 324 (88.5) | |
| *Daily* | 227 | 24 (10.6) | 203 (89.4) | |
| Physical activity during pregnancy | | | | |
| *Never* | 1287 | 149 (11.6) | 1138 (88.4) | 0.643 |
| *1–2 times/week* | 543 | 53 (9.8) | 490 (90.2) | |
| *3–5 times/week* | 209 | 26 (12.4) | 183 (87.6) | |
| *Daily* | 340 | 37 (10.9) | 303 (89.1) | |
| House (rent/owned) | | | | |
| *Owned* | 1963 (79.5) | 212 (10.1) | 1751 (89.2) | 0.498 |
| Worry about upcoming childbirth | 1643 (66.2) | 196 (11.9) | 1447 (88.1) | 0.077 |
| Infertility treatment | 249 (9.9) | 41 (16.5) | 208 (83.5) | 0.004 |
| **Pregnancy and delivery characteristics** | | | | |
| Age, years, mean (SD) | 32.7 (5.57) | 33.8 (5.49) | 32.6 (5.57) | <0.001 |
| Gravida, median (interquartile range) | 4 (3–6) | 5 (3–7) | 4 (3–6) | 0.014 |
| Parity, median (interquartile range) | 3 (2–4) | 3 (2–4) | 3 (2–4) | 0.031 |
| Gestational age at delivery, mean (SD) | 38.7 (2.01) | 34.9 (2.49) | 39.1 (1.35) | <0.001 |
| BMI at delivery, mean (SD) | 31.8 (5.60) | 32.4 (5.72) | 31.8 (5.58) | 0.063 |
| Interpregnancy interval, days, mean (SD) | 1014 (782) | 1098 (884) | 1004 (768) | 0.053 |
| Preexisting hypertension | 56 (2.1) | 21 (37.5) | 35 (62.5) | <0.001 |
| Preexisting diabetes mellitus | 127 (4.7) | 29 (22.8) | 98 (77.2) | <0.001 |
| Previous preterm birth | 1011 (37.3) | 171 (16.9) | 840 (83.1) | <0.001 |
| Previous Caesarian delivery | 876 (32.3) | 158 (18.0) | 718 (82.0) | <0.001 |
| Previous pregnancy loss | 1068 (39.4) | 148 (13.9) | 920 (86.1) | <0.001 |
| Rh antibodies Positive | 2472 (91.2) | 273 (11.0) | 2199 (89.0) | 0.681 |
| Preeclampsia | 93 (3.4) | 35 (37.6) | 58 (62.4) | <0.001 |
| Gestational diabetes mellitus | 889 (32.8) | 105 (11.8) | 784 (88.2) | 0.326 |
| Growth retardation | 19 (0.7) | 2 (10.5) | 17 (89.5) | 0.999* |
| Antepartum hemorrhage | 8 (0.3) | 5 (62.5) | 3 (37.5) | <0.001* |
| Polyhydramnios | 62 (2.3) | 8 (12.9) | 54 (87.1) | 0.622 |
| Oligohydramnios | 58 (2.1) | 20 (34.5) | 38 (65.5) | <0.001 |
| Infection of the amniotic sac | 52 (1.9) | 18 (34.6) | 34 (65.4) | <0.001 |
| Premature rupture of membrane | 378 (13.9) | 55 (14.6) | 323 (85.5) | 0.016 |
| Placental disorders | 28 (1) | 13 (46.4) | 15 (53.6) | <0.001 |
| Placenta Previa | 44 (1.6) | 25 (56.8) | 19 (43.2) | <0.001 |
| Abruption Placentae | 57 (2.1) | 22 (38.6) | 35 (61.4) | <0.001 |
| Streptococcus B carrier | 662 (24.4) | 44 (6.7) | 618 (93.3) | <0.001 |
| Genitourinary infection | 29 (1.1) | 8 (27.6) | 21 (72.4) | 0.004 |

(*Continued*)

**Table 2.** (Continued)

| Risk Factors | Total parous pregnant women | PTB | No PTB | *p*-value |
|---|---|---|---|---|
| Baby Gender | | | | |
| *Male* | 1455 | 180 (12.4) | 1275 (87.6) | 0.012 |
| *Female* | 1252 | 117 (9.4) | 1135 (90.6) | |

Data is number (%) unless otherwise specified.

* Fisher's exact test

SD: standard deviation; PTB: Preterm birth; and BMI: body mass index.

of PTB delivery (0.23). This patient exhibited several risk factors, including oligohydramnios, exposure to passive smoking, a history of previous pregnancy loss, and lower level of education. Notably, there was no history of PTB or CS. Finally, Fig 3(c) illustrates the data of the third parous mother with a higher risk of PTB delivery (0.87) because she had preeclampsia, a history of previous PTB, a history of CS, maternal age > 43 y, and preexisting diabetes. For comparison, the data of the same set of patients were explained using LIME by selecting the ten most important risk factors (S1 Fig).

**Nulliparous women.** We evaluated the performance of six ML classifiers in predicting PTB in nulliparous women. The XGBoost algorithm achieved the highest performance, with an AUC of 0.723 (Fig 4A). Important risk factors for PTB were identified, including maternal BMI at delivery, maternal age, amniotic sac infection, preeclampsia, and history of infertility treatment (Fig 4B and 4C). Other significant risk factors included oligohydramnios, physical activity before pregnancy, and preexisting diabetes.

We further analyzed the individual risk factors associated with a set of nulliparous mothers (Fig 5). For example, patient (a) in Fig 5 had a total risk score of 0.00 and was not diagnosed with PTB. Her most preventive factor was maternal age of 22 y old, no infection of the amniotic sac, BMI at delivery at 28.8 kg/m$^2$, and no premature rupture of membrane. Patient (b) in Fig 5 had a median risk of 0.22 of PTB and was associated with infection of the amniotic sac or membranes (intrauterine infection), preeclampsia, premature rupture of membranes, and a history of infertility treatment. Finally, the nulliparous mother c in Fig 5 had a higher risk of PTB delivery (0.85) owing to her low BMI at delivery (19.9 kg/m$^2$), infection of the amniotic sac, premature rupture of membrane, and no physical activity before pregnancy. SHAP dependence plot for body mass index vs. PBT for nulliparous women showed in S2 Fig display that low BMI had higher SHAP values clarifying the difference between patient (a) and patient (c) in Fig 5. We have also provided LIME for these patients in S3 Fig.

## Discussion

### Principal findings

This study aimed to use ML to develop a prediction model for PTB and identify the key predictors associated with PTB in pregnant women. We used the identified PTB predictors and risk-stratified each patient to generate a risk score using SHAP values for parous and nulliparous mothers. Out of the six most commonly used ML models for PTB prediction, XGBoost exhibited the best performance. The top five most important risk factors in parous women were previous PTB, previous cesarean section, diagnosis of preeclampsia, maternal age, and placenta previa. Among the nulliparous women, the important risk factors were BMI at delivery, maternal age, amniotic fluid infection, premature rupture of membranes, and preeclampsia.

**Table 3. Descriptive characteristics of the nulliparous pregnant women.**

| Risk Factors | Total nulliparous pregnant women | PTB | No PTB | *p*-value |
|---|---|---|---|---|
| **Total** | 801 | 97 | 704 | |
| **Self-reported characteristics** | | | | |
| Planned Pregnancy | 416 (57.7) | 59 (14.2) | 357 (85.8) | 0.029 |
| Employment | 197 (27.4) | 29 (14.7) | 168 (85.3) | 0.212 |
| Education >12 y | 393 (54.6) | 50 (12.7) | 343(87.3) | 0.664 |
| Consanguinity | 193 (4.9) | 24 (12.4) | 169 (87.6) | |
| Passive Smoking | 289 | 31 (10.7) | 258 (89.3) | 0.638 |
| Physical activity before pregnancy | | | | |
| *Never* | 366 | 45 (12.3) | 321 (87.7) | 0.014* |
| *1–2 times/week* | 148 | 25 (16.9) | 123 (83.1) | |
| *3–5 times/week* | 113 | 5 (4.4) | 108 (95.6) | |
| *Daily* | 46 | 6 (13.0) | 40 (87.0) | |
| Physical activity during pregnancy | | | | |
| *Never* | 367 | 50 (13.6) | 317 (86.4) | 0.278* |
| *1–2 times/week* | 179 | 22 (12.3) | 157 (87.7) | |
| *3–5 times/week* | 59 | 5 (8.5) | 54 (91.5) | |
| *Daily* | 79 | 5 (6.3) | 74 (93.7) | |
| House (rent/owned) | | | | |
| *Owned* | 582 (82.2) | 70 (12.0) | 512 (88.0) | 0.834 |
| Worry about upcoming childbirth | 521 (73.5) | 70 (13.4) | 451 (86.6) | 0.116 |
| Infertility treatment | 65 (9.2) | 16 (24.6) | 49 (75.4) | 0.001 |
| **Pregnancy and delivery characteristics** | | | | |
| Age, years, mean (SD) | 27.6 (6.00) | 28.5 (6.52) | 27.5 (5.92) | 0.119 |
| Gestational age at delivery, mean (SD) | 38.7 (2.41) | 34.1 (3.29) | 39.4 (1.31) | <0.001 |
| BMI at delivery, mean (SD) | 30.8 (6.99) | 30.6 (14.28) | 30.9 (5.29) | 0.739 |
| Preexisting hypertension | 13 (1.6) | 4 (30.8) | 9 (69.2) | 0.061 |
| Preexisting diabetes mellitus | 12 (1.5) | 5 (41.7) | 7 (58.3) | 0.009* |
| Rh antibodies Positive | 734 (91.6) | 91 (12.4) | 643 (87.6) | 0.408 |
| Preeclampsia | 50 (6.2) | 16 (32.0) | 34 (68.0) | <0.001 |
| Gestational diabetes mellitus | 201(25.1) | 23 (11.4) | 178 (88.6) | 0.738 |
| Growth retardation | 14 (1.7) | 2 (14.3) | 12 (85.7) | 0.638* |
| Antepartum hemorrhage | 3 (0.4) | 2 (66.7) | 1 (33.3) | 0.040 |
| Polyhydramnios | 20 (2.5) | 3 (15.0) | 17 (85.0) | 0.724* |
| Oligohydramnios | 18 (2.2) | 6 (33.3) | 12 (66.7) | 0.015* |
| Infection of the amniotic sac | 34 (4.2) | 14 (41.2) | 20 (58.8) | <0.001 |
| Premature rupture of membrane | 200 (24.9) | 34 (17.0) | 166 (83.0) | 0.014 |
| Placental disorders | 4 (0.5) | 1 (25.0) | 3 (75.0) | 0.404* |
| Placenta Previa | 4 (0.5) | 1 (25.0) | 3 (75.0) | 0.404* |
| Abruption Placentae | 14 (1.7) | 6 (42.9) | 8 (57.1) | <0.001 |
| Streptococcus B carrier | 227 (28.3) | 23 (10.1) | 204 (89.9) | 0.281 |
| Genitourinary infection | 9 (1.1) | 1 (11.1) | 8 (88.9) | 0.999* |
| Baby Gender | | | | |
| *Male* | 423 | 49 (11.6) | 374 (88.4) | 0.701 |
| *Female* | 377 | 47 (12.5) | 330 (87.5) | |

Data is number (%) unless otherwise specified.

* Fisher's exact test

SD: standard deviation; BMI: body mass index; and PTB: Preterm birth.

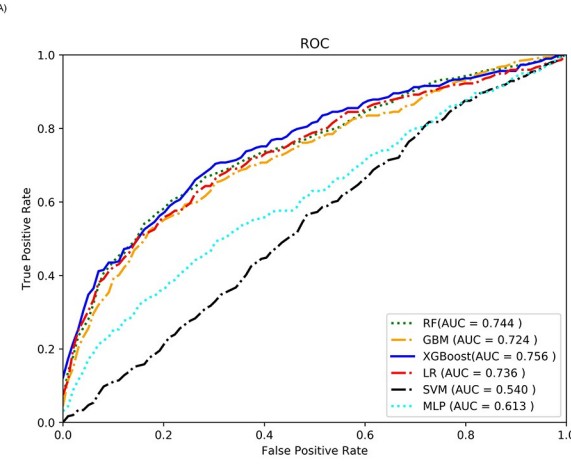

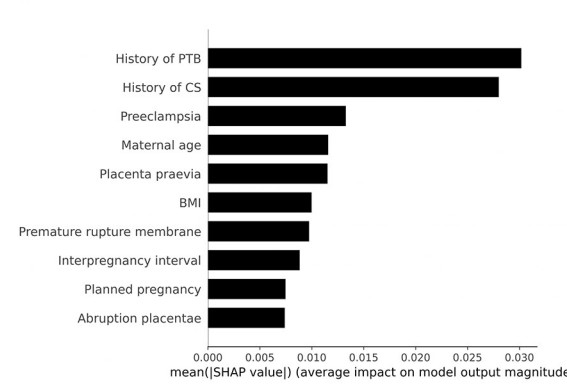

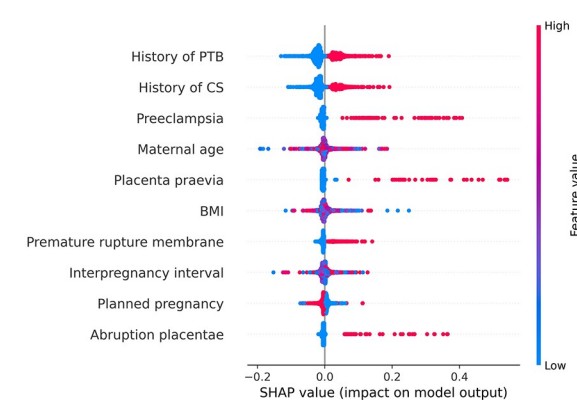

**Fig 2.** A. ROC curve for PTB prediction in parous women (n = 2708). B. SHAP-based feature importance plot for parous women. C. Summary plot for top 10 SHAP-based risk factors in parous women. Each dot in the graph indicates a patient and her relative risk towards PTB prediction. Several patients at the same point create a dense region. The colors indicate the feature values on the right side (vertically): blue indicates lower values while red indicates higher values of a risk factor. For instance, for previous cesarean section (CS) delivery, when the number of CS deliveries increases then the risk of PTB delivery increases while patients with lower (or no CS) deliveries are at a lower risk of PTB. We also observed negative interactions, such as patients with higher BMI, are at a relatively lower risk of PTB, whereas those with lower BMI are at a higher risk.

Individual patient risk scores were calculated to determine the level of risk. These results were internally validated. The results of both the SHAP and LIME algorithms aligned, except in a few cases, mainly because LIME does not guarantee an accurate distribution of the effects [21, 42]. However, because SHAP focuses on local accuracy and consistency, it generates more accurate model outcomes. Both the SHAP and LIME guide clinicians toward individual risk categorization. This will prevent over investigation and undue economic burdens on health systems.

### Results in the context of what is known

PTB rates vary by region and country. The overall PTB rate in this study was 11.23%. North America recorded a rate of 11.2%, with the highest rates in North Africa and Sub-Saharan Africa at 13.4 and 12%, respectively [44]. The US ranked among the top ten countries for PTB

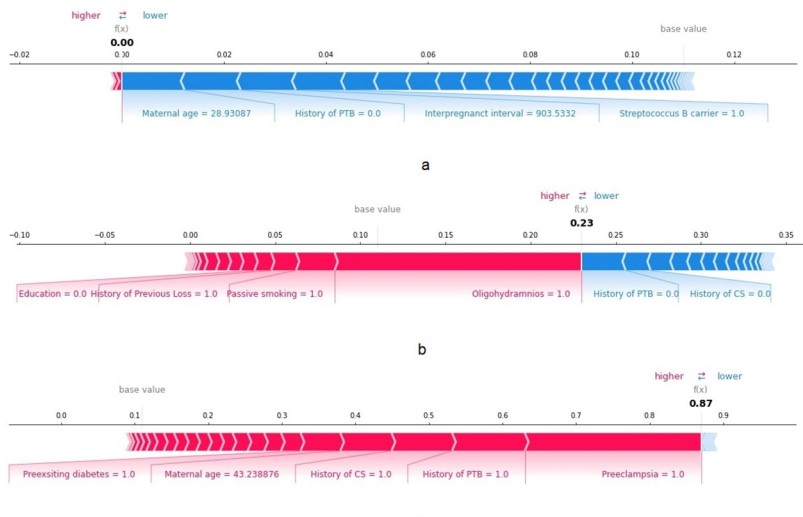

**Fig 3. Individual patient set analysis for parous women using SHAP.** The feature values in red indicate the risk factors increasing the chances of PTB, whereas those in blue indicate factors reducing the chances of PTB. The size of the risk factor indicates its degree of influence on that specific patient. Patient (a) is at lower risk, patient (b) median risk, and patient (c) higher risk of PTB.

in 2014 at 9.6%. Interestingly, the other nine countries are developing countries with struggling economies and disparities in health services [44].

The strength of AI and ML is their ability to learn from new inputs and leverage those insights to enhance health outcomes and patient experiences. This model, XGBoost, achieved an AUC value of 0.735 and identified plausible and well-known risk factors for clinicians, such as previous cesarean section, previous PTB, diagnosis of preeclampsia, maternal age, placenta previa, and BMI [45]. Despite, Lee et al. [46] achieving an AUC of 0.54–0.83, the extensive list of predictors described was inexplicable and unrelated to predictors for PTB (e.g., "upper gastrointestinal tract symptom, gastroesophageal reflux disease, Helicobacter pylori"; all entities describing the same symptom) and particularly for practicing clinicians. Although Sun et al. [16] achieved a maximum AUC of 0.885, they recognized risk factors, such as age, magnesium, fundal height, serum inorganic phosphorus, mean platelet volume, waist size, total cholesterol, triglycerides, globulins, and total bilirubin for their prediction model. Risk factors, such as fundal height and waist size, were determined by healthcare professionals. Because these measurements are skill-dependent, they may be inaccurate, resulting in misleading outcomes, particularly in obese patients [47]. A study using retrospective medical data reported an AUC of 0.739 [16]. The risk factors included blood pressure, blood glucose, lipids, and uric acid as metabolic predictors of PTB. Performing uric acid tests is not routine in all pregnant women. Although it is a significant risk factor, it has only been measured for pregnancy outcomes in women with preeclampsia/eclampsia [48]. In a systematic review of traditional prediction models for the risk of spontaneous PTB and based on routine clinical parameters, the AUC for these models ranged from 0.54 to 0.67 with consequential outcomes that our ML predictions are beginning to demonstrate potential [49]. Moreover, unlike LR, the XGBoost algorithm uses a nonparametric assessment; therefore, the correlation of the independent variables has no significant impact on the weighted ranking of each variable. Significantly, XGBoost demonstrated highly promising performance in patients with diabetic retinopathy, with an AUC of 0.99 [50]. The difference in the AUC from our results may be explained by the sample size of

(A)

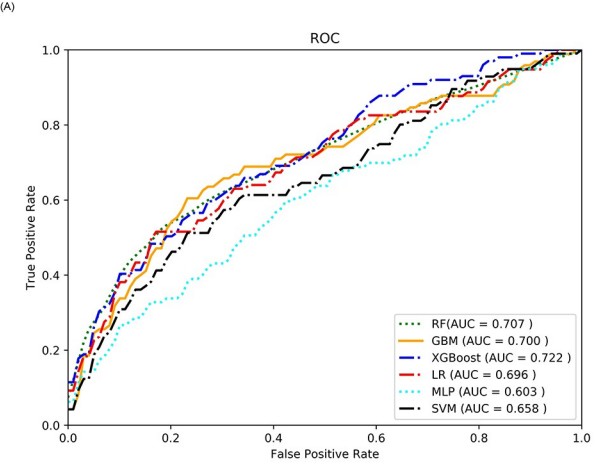

(B)

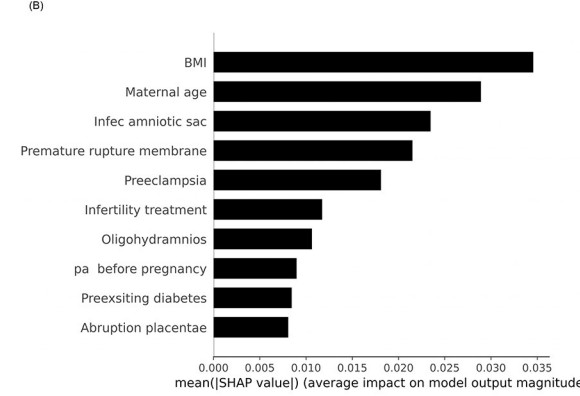

(C)

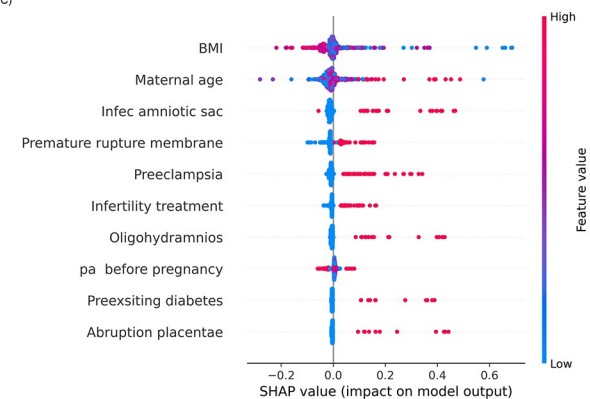

**Fig 4.** A. ROC curve for PTB prediction in the nulliparous women (n = 810). B. SHAP-based feature importance plot for nulliparous women. C. Summary plot for top 10 SHAP-based risk factors in nulliparous women.

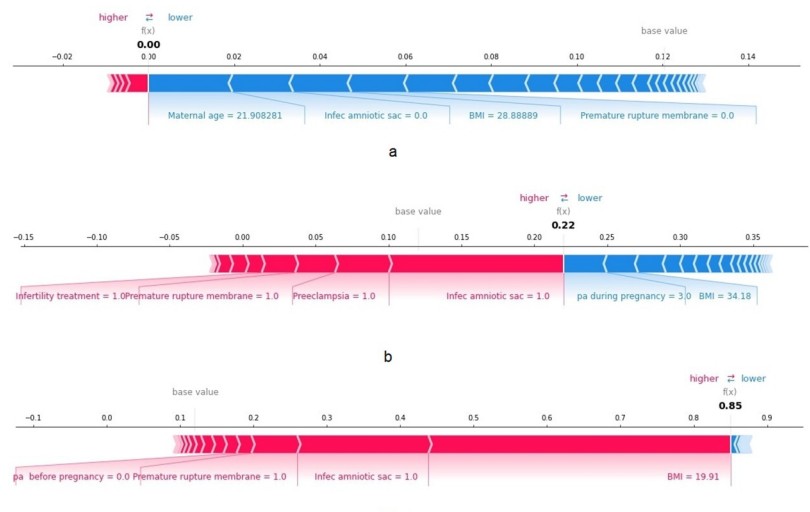

**Fig 5. Individual patient set analysis for nulliparous women using SHAP.** Patient (a) is at lower risk, patient (b) median risk, and patient (c) higher risk of PTB.

32 452 women [50]. This suggests that XGBoost can also be used as a predictive model for other diseases [51]. Finally, our study progressed beyond the black-box approach (Table 1), in which the results could be easily interpreted by clinicians, thereby enhancing the clinical usability of this study.

## Clinical implications

ML in healthcare increases the accuracy of prediction and diagnosis, thus helping clinicians make informed decisions and personalize patient health care. Our study shaped the predictive probabilities of PTB in asymptomatic, nulliparous, and at-risk mothers using real-world data. Nulliparous women often present a dilemma for clinicians, even when traditional prediction models are used. The results confirmed that traditional models discriminated poorly for nulliparous women (AUC 0.51–0.56) [49]. We produced individual risk categorizations for PTB and, more importantly, for nulliparous women using SHAP values, where each risk factor was assigned a weight by the algorithm. Once patients are deemed at-risk or high-risk, their treating physicians will then follow-up patients using serial endovaginal ultrasound measurements of cervical length from 16 weeks of gestation to 24 weeks of gestation [52]. Thus, the SHAP method is a stage closer to a reliable method for making the output of the XGBoost model clinically interpretable.

## Research implications

The set of risk factors used in this study to derive the ML model provides the first step, and other indicators, such as ultrasound parameters, biomarkers, and fetal fibronectin, can be incrementally added to test the performance of the XGBoost method for improved predictability. Our results must be externally validated and verified in other populations, particularly asymptomatic and nulliparous women. The development of a risk calculator from these predictors to stratify risk based on the scores obtained would make an immense contribution to clinicians [50].

## Strengths and limitations

In this study, risk factor selection to build the ML model for PTB prediction was based on the medical literature. The major advantage of this study was personalized risk stratification with SHAP values and easy clinical interpretability, fostering individual management recommendations. XGBoost outperformed the five other ML techniques with the highest AUC value (0.735), which was the key guide for evaluating the function of the predictive model. In addition, this algorithm is less time-consuming than other ML algorithms.

This study has several limitations. One of the limitations of this study is the absence of information regarding the etiology of PTB in our population, specifically whether it was indicated or spontaneous. Consequently, we restricted our analysis to parous and nulliparous women.

Second, the results are internally validated. We must be cautious about generalizability, as the model needs to be tested on other datasets and evaluated in other centers. Although XGBoost has considerable potential, the model performance can be improved by adding more indicators. This study provides only a preliminary explanation of the interpretability of the ML model.

## Conclusion

This study highlights the use of a novel technology (XGBoost) for risk stratification among individual pregnant women, particularly asymptomatic, nulliparous, and at-risk patients with

PTB, and uses the SHAP method to clinically interpret the model's outputs, thus enhancing the clinician's ability to detect risks earlier. This model may be used as a screening tool to aid clinicians in understanding the impact of each risk factor on each patient, thus guiding clinicians in administering appropriate care to reduce morbidity and mortality related to PTB.

## Supporting information

**S1 Table. Features selection from the literature of potentials risk factors of preterm birth.** (DOCX)

**S2 Table. Selected variables for prediction of preterm birth.** (DOCX)

**S1 File. Definition of SHAP and LIME algorithms.** (DOCX)

**S1 Fig. Individual patient set analysis using LIME in parous women.** LIME prediction contains two parts: the left-side probability of the patient having PTB, and the right-side local explanation of the risk factors for a specific patient. Patient (a) is at a lower risk, patient (b) is at a median risk, and patient (c) is at a higher risk of PTB. (TIF)

**S2 Fig. SHAP dependence plot for BMI versus preterm birth for nulliparous women.** (TIF)

**S3 Fig. Individual patient set analysis using LIME in nulliparous women.** LIME prediction contains two parts: the left-side probability of the patient having PTB, and the right-side local explanation of the risk factors for a specific patient. Patient (a) is at a lower risk, patient (b) is at a median risk, and patient (c) is at a higher risk of PTB. (TIF)

## Author Contributions

**Conceptualization:** Wasif Khan, Nadirah Ghenimi, Amir Ahmad.

**Data curation:** Wasif Khan, Nadirah Ghenimi, Luai A. Ahmed.

**Formal analysis:** Wasif Khan.

**Methodology:** Wasif Khan, Amir Ahmad, Luai A. Ahmed.

**Resources:** Nadirah Ghenimi.

**Software:** Wasif Khan, Luai A. Ahmed.

**Supervision:** Nazar Zaki, Amir Ahmad, Mohammad M. Masud, Romona Govender, Luai A. Ahmed.

**Validation:** Wasif Khan, Luai A. Ahmed.

**Writing – original draft:** Nadirah Ghenimi.

**Writing – review & editing:** Nazar Zaki, Nadirah Ghenimi, Amir Ahmad, Jiang Bian, Mohammad M. Masud, Nasloon Ali, Romona Govender, Luai A. Ahmed.

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
