## [Decision Letter · Decision Letter 0]

19 Jun 2023

PONE-D-23-12563Predicting preterm birth using explainable machine learning in a prospective cohort of pregnant womenPLOS ONE

Dear Dr. Ghenimi,

Thank you for submitting your manuscript to PLOS ONE. After careful consideration, we feel that it has merit but does not fully meet PLOS ONE’s publication criteria as it currently stands. Therefore, we invite you to submit a revised version of the manuscript that addresses the points raised during the review process.

ACADEMIC EDITOR:

1) “Preterm birth (PTB) is a complex condition with multifactorial etiology.“

2) “e.g., advanced maternal age, low pre-pregnancy weight)” Both being very thin and being overweight are risk factors. Correct as: adolescent pregnancy, advanced maternal age….

3) Include the long version of the text in the first abbreviation. For example “ML models”

4) Add all table abbreviations below the table.

5) “This analysis included 3509” Split it into paragraphs.

6) Please check the text and tables for grammar, seek professional grammar service if necessary!

We look forward to receiving your revised manuscript.

Kind regards,

Burak Bayraktar

Academic Editor

PLOS ONE

3. We note that Figure 1 in your submission contain copyrighted images. All PLOS content is published under the Creative Commons Attribution License (CC BY 4.0), which means that the manuscript, images, and Supporting Information files will be freely available online, and any third party is permitted to access, download, copy, distribute, and use these materials in any way, even commercially, with proper attribution. For more information, see our copyright guidelines: http://journals.plos.org/plosone/s/licenses-and-copyright.

b.If you are unable to obtain permission from the original copyright holder to publish these figures under the CC BY 4.0 license or if the copyright holder’s requirements are incompatible with the CC BY 4.0 license, please either i) remove the figure or ii) supply a replacement figure that complies with the CC BY 4.0 license. Please check copyright information on all replacement figures and update the figure caption with source information. If applicable, please specify in the figure caption text when a figure is similar but not identical to the original image and is therefore for illustrative purposes only.

Additional Editor Comments (if provided):

1) “Preterm birth (PTB) is a complex condition with multifactorial etiology.“

2) “e.g., advanced maternal age, low pre-pregnancy weight)” Both being very thin and being overweight are risk factors. Correct as: adolescent pregnancy, advanced maternal age….

3) Include the long version of the text in the first abbreviation. For example “ML models”

4) Add all table abbreviations below the table.

5) “This analysis included 3509” Split it into paragraphs.

6) Please check the text and tables for grammar, seek professional grammar service if necessary!

Reviewers' comments:

Reviewer's Responses to Questions

**Comments to the Author**

1. Is the manuscript technically sound, and do the data support the conclusions?

Reviewer #1: Partly

Reviewer #2: Yes

2. Has the statistical analysis been performed appropriately and rigorously? 

Reviewer #1: Yes

Reviewer #2: Yes

3. Have the authors made all data underlying the findings in their manuscript fully available?

Reviewer #1: No

Reviewer #2: No

4. Is the manuscript presented in an intelligible fashion and written in standard English?

Reviewer #1: Yes

Reviewer #2: Yes

5. Review Comments to the Author

Reviewer #1: I am really grateful for reviewing this manuscript. In my opinion, this manuscript can be published once some revision is done successfully. This study used 3509 pregnant women, six machine learning models and achieved the area under the curve of 72.3%-73.5% (boosting) for the prediction of preterm birth. This study employed SHAP (SHapley Additive exPlanations) as an explainable artificial intelligence approach. I would argue that this is a great achievement. However, it can be noted that the direction of association between body mass index and preterm birth is not very clear in the SHAP summary plot for nulliparious pregnant women. I would like to suggest the authors to derive the SHAP dependence plot for body mass index vs. preterm birth for nulliparous women, give that this would aid in clarifying the issue.

Reviewer #2: The manuscript describes the use of explainable machine learning to predict preterm birth through medical records and self-administered questionnaires. The authors separate nulliparous from multiparous. This must be declared in the title, for example, by adding ... "in nulliparous and multiporous women."

The manuscript is easy to read.

Other comments:

Table 1, described in Introduction, could be moved to the Result Section since it is part of the study.

The experimental setting provides little information due to everything explained before. Therefore, it may be better to improve this section and move it to Result Section.

The tables are too big; in dichotomic responses, put only one.

Which are the difference between Gravida and Parity?

These are discrete quantitative variables, so the median and range are recommended.

The authors describe LIME approximation, but all the results are in the supplementary figures and tables. Therefore, it is recommended to reduce LIME's description of methodology since the authors do not give the importance expected in the manuscript.

In line 185, it is described as a different PTB with no PTB on maternal age, gravida, ... but it does not specify which characteristics are present in PTB. In line 193, it is the same question.

In line 209, was BMI a qualitative or quantitative variable?

In line 211, it is unclear if the author put a patient as an example, as in (b) and (c).

6. PLOS authors have the option to publish the peer review history of their article (what does this mean?). If published, this will include your full peer review and any attached files.

Reviewer #1: No

Reviewer #2: No

---

## [Author Response · Author response to Decision Letter 0]

3 Aug 2023

Reviewers' Comments:

1. The SHAP dependence plot for BMI vs. preterm birth for nulliparous women has been added as a supplementary file (S2 Fig) to clarify the association (Line 307-309).

2. The title has been updated to include "nulliparous and multiparous women" (Line 136).

3. Table 1 has been moved to the Methods section (Line 137).

4. The experimental setting has been improved and detailed in the Methods Section (Lines 174-179).

5. Table 2 and Table 3 have been revised to present dichotomic responses more clearly.

6. Definitions and differences between "gravida" and "parity":

Gravida represents the total number of pregnancies a woman has had, while parity specifically refers to the number of pregnancies in which a woman has given birth to viable infants.

7. The description of the LIME methodology has been reduced, as recommended (Lines 169-171 deleted).

8. The characteristics of PTB and non-PTB women have been included in the results.

9. Clarifications have been made regarding the patient examples in Fig 3 and Fig 5 (a, b, c).

Reviewer #1: I am really grateful for reviewing this manuscript. In my opinion, this manuscript can be published once some revision is done successfully. This study used 3509 pregnant women, six machine learning models and achieved the area under the curve of 72.3%-73.5% (boosting) for the prediction of preterm birth. This study employed SHAP (SHapley Additive exPlanations) as an explainable artificial intelligence approach. I would argue that this is a great achievement. However, it can be noted that the direction of association between body mass index and preterm birth is not very clear in the SHAP summary plot for nulliparous pregnant women. I would like to suggest the authors to derive the SHAP dependence plot for body mass index vs. preterm birth for nulliparous women, give that this would aid in clarifying the issue.

Answer: Thank you for your suggestion. The plot depicting the SHAP dependence plot for BMI was submitted as a supplementary file (S2 Fig). The figures displaying the low BMIs with higher SHAP values, clarify the differences between patient (a) and patient (c) in Fig 5 (L307-L309).

Reviewer #2: The manuscript describes the use of explainable machine learning to predict preterm birth through medical records and self-administered questionnaires. The authors separate nulliparous from multiparous. This must be declared in the title, for example, by adding ... "in nulliparous and multiparous women."

Answer: Thank you for this invaluable suggestion. We edited the title to include nulliparous, “Predicting preterm birth using explainable machine learning in a prospective cohort of nulliparous and multiparous women.”

The manuscript is easy to read.

Other comments:

Table 1, described in Introduction, could be moved to the Result Section since it is part of the study.

Answer: Table 1 was moved into the Methods section as it is not described in Results (L137).

The experimental setting provides little information due to everything explained before. Therefore, it may be better to improve this section and move it to Result Section.

Answer: The experimental setting was modified to include a detailed description (L174-L179).

The tables are too big; in dichotomic responses, put only one.

Answer: We agree with the reviewer. Table 2 and Table 3 were edited and the “No” was deleted. As the totals vary for the different variables, we added percentages in brackets for the “Yes”. 

Which is the difference between Gravida and Parity?

Answer: Definitions and differences between "gravida" and "parity" in the context of obstetrics:

Gravida: Gravida refers to the number of times a woman has been pregnant, regardless of the outcome of the pregnancy. It includes both ongoing pregnancies and pregnancies that have ended in any outcome (live births, stillbirths, miscarriages, ectopic pregnancies, etc.). Gravida is a term used to indicate the total number of pregnancies a woman has experienced.

Reference: Cunningham, F. G., Leveno, K. J., Bloom, S. L., Spong, C. Y., Dashe, J. S., & Hoffman, B. L. (2018). Williams obstetrics. McGraw-Hill Education.

Parity: Parity refers to the number of pregnancies in which the foetus or foetuses have reached the stage of viability (at least 20 weeks of gestation) and have been born alive, regardless of whether the child is currently alive or deceased. It does not include ongoing pregnancies, stillbirths, or pregnancies that ended before 20 weeks. Parity is a term used to indicate the number of times a woman has given birth to a viable infant.

Reference: American College of Obstetricians and Gynecologists. (2018). ACOG Committee Opinion No. 736: Optimizing Postpartum Care. Obstetrics and Gynecology, 131(5), e140-e150.

In summary, gravida represents the total number of pregnancies a woman has had, while parity specifically refers to the number of pregnancies in which a woman has given birth to viable infants.

These are discrete quantitative variables, so the median and range are recommended.

Answer: Thank you for this suggestion and the authors agree that the variables, “gravida and parity” are discrete variables. The median and range was added to Table 2 (L210).

The authors describe LIME approximation, but all the results are in the supplementary figures and tables. Therefore, it is recommended to reduce LIME's description of methodology since the authors do not give the importance expected in the manuscript.

Answer: The authors agree and the LIME description in the methodology was edited as required (L169-L171 deleted).

In line 185, it is described as a different PTB with no PTB on maternal age, gravida, ... but it does not specify which characteristics are present in PTB. In line 193, it is the same question.

 Answer: We added the characteristics of PTB women and no PTB women in the results.

The more significant characteristics for women who had experienced PTB is maternal age, gravida and, level of education passive smoking, infertility treatment, pre-existing hypertension and pre-existent Diabetes. `

In line 209, was BMI a qualitative or quantitative variable?

Answer : The BMI is a quantitative variable.

In line 211, it is unclear if the author put a patient as an example, as in (b) and (c).

Answer: Yes, we did use a patient in Fig 3 and Fig 5 a, b and c. These are 3 different patients described in the examples. We addressed this suggestion and edited this in the manuscript. The authors hope that this is now clearer to the reviewer.

---

## [Editor Report · Decision Letter 1]

24 Sep 2023

PONE-D-23-12563R1Predicting preterm birth using explainable machine learning in a prospective cohort of nulliparous and multiparous pregnant womenPLOS ONE

Dear Dr. Ghenimi,

Thank you for submitting your manuscript to PLOS ONE. After careful consideration, we feel that it has merit but does not fully meet PLOS ONE’s publication criteria as it currently stands. Therefore, we invite you to submit a revised version of the manuscript that addresses the points raised during the review process.

We look forward to receiving your revised manuscript.

Kind regards,

Burak Bayraktar

Academic Editor

PLOS ONE

Journal Requirements:

Additional Editor Comments:

The authors made various edits. However, there are still grammatical errors. I'm giving some of it.

1) “infection of the amniotic” It's not an accurate definition. Correct as “intra-amniotic infection” or “infection of the amniotic fluid” or “chorioamnionitis”.

2) Key words should not contain abbreviations.

3) “models lack proper interpretations for clinicians.” Correct as: “models lack proper interpretation for clinicians.”

4) “Preterm birth (PTB) complications is” Correct as: “Preterm birth (PTB) complications are”

5) “Hence, to be useful,“ Correct as: “Hence, for them to be useful”

6) Correct as: “with PTB were significantly different from”

7) Correct as: “infection of the amniotic sac”

8) Correct as: “and a low level of education. ”

9) Correct as: “and a history of previous PTB are”

10) Correct as: “conditions such as preexisting diabetes”

11) Correct as: “infection of the amniotic sac,”

12) Correct as: “patient (b) is at median risk”

13) Correct as: “risk of 0.22 for PTB”

14) Correct as: “factors in parous women were having a previous PTB”

15) Correct as: “maternal age, infection of the amniotic fluid,”

---

## [Author Response · Author response to Decision Letter 1]

15 Oct 2023

Subject: Response to Editor for Submission of Manuscript Titled 

"Predicting Preterm Birth using Explainable Machine Learning in a Prospective Cohort of Pregnant Women" to PLOS ONE

15 October 2023

Dear Editor,

Thank you for the comprehensive feedback on our manuscript submitted to PLOS ONE. We appreciate the time and effort invested by the reviewers and the editorial team to evaluate our work. I acknowledge all the comments and have made the necessary revisions to address them. In the initial review process, minor revisions were requested, and we are pleased to report that we have successfully incorporated these changes.

In response to your feedback, we've engaged professional language and grammar editors to meticulously correct and enhance the quality, punctuation, and spelling in our manuscript. We've attached a certificate from this editing service for your reference.

Additionally, while revising our submission, we followed PLOS ONE's guidelines and uploaded our figure files to the Preflight Analysis and Conversion Engine (PACE) digital diagnostic tool. 

The revised manuscript (with tracked changes), the clean manuscript, and this response letter have been uploaded to the submission portal as instructed.

With these revisions, we believe the manuscript now aligns with PLOS ONE's high standards. We appreciate the opportunity to improve our work and are eager for your feedback on the revised version.

Below is a point-by-point response to the comments:

Editor's Comments and Author's Responses:

1) Comment: “Infection of the amniotic” It's not an accurate definition. Correct as “intra-amniotic infection” or “infection of the amniotic fluid” or “chorioamnionitis”.

Response: We have corrected this phrase to "infection of the amniotic sac" throughout the manuscript. We used ICD10 code for a selection of the features from the Electronic medical records and the ICD10 code for intraamniotic infection is: 

` 2024 ICD-10-CM Diagnosis Code O41.1090: `Infection of amniotic sac and membranes, unspecified, unspecified trimester, not applicable or unspecified`

2) Comment: Key words should not contain abbreviations.

Response: We have removed all abbreviations from the keywords. L 60 : Keywords: Preterm birth prediction, Machine Learning model, risk assessment, SHapley Additive exPlanations, Local interpretable model-agnostic explanations. 

3) Comment: “models lack proper interpretations for clinicians.” Correct as: “models lack proper interpretation for clinicians.”

Response: We have made the suggested correction: L 28 the sentence was re-edited with ` While machine learning (ML) algorithms have shown promise in PTB prediction, the lack of interpretability in existing models hinders their clinical utility

4) Comment: “Preterm birth (PTB) complications is” Correct as: “Preterm birth (PTB) complications are”

Response: We have corrected this phrase as suggested.

5) Comment: “Hence, to be useful,“ Correct as: “Hence, for them to be useful”

Response: This phrase has been revised accordingly.

6) Comment: Correct as: “with PTB were significantly different from”

Response: The necessary changes have been made.

7-8) Comment: Multiple corrections suggesting usage of “infection of the amniotic sac” and "and a low level of education."

Response: These corrections have been applied in the manuscript as suggested.

9-15) Comment: Several phrasing corrections were provided. 

Response: All the recommended phrasing corrections have been applied throughout the manuscript.

---

## [Editor Report · Decision Letter 2]

23 Oct 2023

Predicting preterm birth using explainable machine learning in a prospective cohort of nulliparous and multiparous pregnant women

PONE-D-23-12563R2

Dear Dr. Ghenimi,

We’re pleased to inform you that your manuscript has been judged scientifically suitable for publication and will be formally accepted for publication once it meets all outstanding technical requirements.

Kind regards,

Burak Bayraktar

Academic Editor

PLOS ONE
---

## [Editor Report · Acceptance letter]

27 Oct 2023

PONE-D-23-12563R2 

Predicting preterm birth using explainable machine learning in a prospective cohort of nulliparous and multiparous pregnant women 

Dear Dr. Ghenimi:

I'm pleased to inform you that your manuscript has been deemed suitable for publication in PLOS ONE. Congratulations! Your manuscript is now with our production department. 

Kind regards, 

on behalf of

Dr. Burak Bayraktar 

Academic Editor

PLOS ONE